# Neuro-evolutionary Continual Reinforcement Learning

**Pengyi Li**[1]  **Hongyao Tang**[1]  **Yifu Yuan**[1]  **Yan Zheng**[1]  **Xin Xu**[2]  **Jianye Hao**[1]

## Abstract

Deploying robots in open-ended real-world environments demands continual learning capabilities to adapt to an ever-expanding range of tasks. This requires retaining previously acquired skills without forgetting while effectively leveraging prior knowledge to learn new ones. Inspired by neuroscience, we propose **N**euro-**evol**utionary **C**ontinual **R**einforcement **L**earning (**Nevo-CRL**). Nevo-CRL maintains a fixed-capacity monolithic policy network, solving tasks by optimizing inter-layer connectivity and neuron parameters. For each new task, Nevo-CRL constructs a mask population to selectively activate the outputs of each hidden layer, thereby forming a task-specific policy population. Upon completing each task, the best-performing mask is stored, and its activated neurons are frozen to prevent catastrophic forgetting. To facilitate knowledge transfer, Nevo-CRL reuses neurons from acquired skills based on semantic similarity between tasks, while dynamically allocating additional neurons for task-specific adaptation. In the learning process, Nevo-CRL iteratively adjusts masks via importance-guided crossover to optimize the policy network connectivity. To improve neuron utilization, we prune low-activity connections to recycle neurons. Experiments demonstrate that Nevo-CRL achieves state-of-the-art performance among continual RL methods. The code is available at https://github.com/yeshenpy/Nevo-CRL.

## 1. Introduction

Reinforcement Learning (RL) (Sutton & Barto, 1998) has achieved impressive progress across a wide range of do-

[1]School of Computer Software, Tianjin University, China [2]College of Intelligence Science, National University of Defense Technology, China. Correspondence to: Hongyao Tang <tanghongyao@tju.edu.cn>, Jianye Hao <jianye.hao@tju.edu.cn>.

*Proceedings of the 43$^{rd}$ International Conference on Machine Learning*, Seoul, South Korea. PMLR 306, 2026. Copyright 2026 by the author(s).

mains, including robotic control (Johannink et al., 2019; Singh et al., 2022; Liu et al., 2021; Nguyen & La, 2019), game AI (Silver et al., 2016; Vinyals et al., 2019; Berner et al., 2019), recommender systems (Zou et al., 2019; Afsar et al., 2023; Xin et al., 2022) and chip design (Li et al., 2026a). Powered by deep neural networks, RL can learn complex behavioral policies through trial-and-error interactions with the environment. However, when faced with a sequence of tasks, RL suffers from several challenges (Khetarpal et al., 2022; Wołczyk et al., 2021), including catastrophic forgetting (French, 1993; Bengio et al., 2020), negative transfer across tasks (Zhang et al., 2023; Jiang et al., 2023), and plasticity loss (Nikishin et al., 2023; Abbas et al., 2023; Dohare et al., 2024). These problems severely limit its deployment in many real-world scenarios. For example, a household robot must continuously optimize its policy to handle an evolving set of daily tasks (Ayub et al., 2024). As a result, how to generalize across previously learned tasks, avoid forgetting, and quickly adapt to new, unseen tasks has become a key research problem—commonly known as *continual learning* or *lifelong learning* (Mendez & Eaton, 2023; Abel et al., 2023; Fu et al., 2022).

The human brain, shaped by thousands of years of evolution, stands as one of the most successful examples of continual learning. From birth, it continuously acquires a wide range of skills to adapt to the complexity of the physical world (Hassabis et al., 2017; Wickramasinghe et al., 2023). Therefore, understanding how the brain addresses an ever-increasing number of new tasks may provide valuable insights for overcoming the aforementioned challenges (de Ven et al., 2020). We summarize four key characteristics as follows:

❶ **Sparse Connectivity**: When facing new tasks, only a small subset of neurons in the human brain are activated in response to the given stimulus. The brain does not sacrifice sparsity for knowledge accumulation; instead, it constructs more efficient connectivity pathways through rewiring (Babadi & Sompolinsky, 2014; Gurbuz & Dovrolis, 2022). *This suggests that maintaining sparsity is essential in continual learning* (Hadsell et al., 2020).

❷ **Learning and Evolution**: The human brain learns by adjusting the strength of synaptic signals between neurons (Park et al., 2014; Cooke & Bliss, 2006). In addition,

it continuously eliminates or forms synaptic connections to restructure neural circuits for better task adaptation (Fu & Zuo, 2011; Kasai et al., 2010; Deger et al., 2012). Inspired by this, *continual optimization of both parameters and connectivity is essential in policy learning.*

❸ **Neural Stability**: After completing the learning of a task, the brain ensures the stability of input signals to the relevant synapses and stores them over the long term (Grutzendler et al., 2002; Parisi et al., 2019). When facing new tasks, these sets of synapses remain unchanged (Zuo et al., 2005; Yang et al., 2009; 2014). Therefore, *it is essential to maintain the parameters and connectivity associated with previously learned tasks to prevent forgetting.*

❹ **Fixed Capacity**: The number of neurons in the human brain remains largely unchanged throughout life (Ming & Song, 2011; Gurbuz & Dovrolis, 2022). This indicates that, *rather than introducing new neurons to handle new tasks, a continual learning system may effectively solve them by reusing existing ones within a fixed-capacity architecture.*

Inspired by the above neuroscientific principles, we propose the **Neuro-evo**lutionary **C**ontinual **R**einforcement **L**earning framework (**Nevo-CRL**). Nevo-CRL is built upon a fixed-capacity monolithic policy network and employs sparse masks to form task-specific network connectivity to learn new tasks (❹ Fixed Capacity and ❶ Sparse Connectivity). To construct the mask, Nevo-CRL reuses previously learned neurons based on task semantic similarity for efficient knowledge transfer and dynamically allocates additional new neurons for plasticity. To solve the new task, Nevo-CRL maintains a population of masks. We subsequently replicate the monolithic policy and pair each copy with a corresponding mask to construct task-specific policies. This replication ensures that parameter updates for each policy remain isolated, preventing interference during optimization. These policies then interact with the environment, and the collected experiences are stored in a shared replay buffer for optimization and learning. (❷ Parameter Learning). The policy performance is used as fitness to evaluate the quality of its corresponding mask. The population undergoes periodic mask evolution, where we introduce the bit-level importance-guided crossover method that focuses on the more critical mask bits to enhance evolutionary efficiency (❷ Connection Evolution). Additionally, Nevo-CRL performs network pruning during training by removing low-contributing connections, thereby improving parameter efficiency. To prevent forgetting of previously learned tasks, Nevo-CRL freezes the parameters and connections associated with prior tasks when learning new ones, ensuring that acquired skills remain unaffected (❸ Neural Stability). Our experiments on the Continual World benchmark (Wołczyk et al., 2021) demonstrate that Nevo-CRL achieves the best average performance, substantially reduces forgetting, and provides competitive generalization compared with strong baselines.

We summarize our contributions as follows: 1) Inspired by neuroscience, we propose the Neuro-evolutionary Continual Reinforcement Learning framework Nevo-CRL, which mitigates forgetting while achieving efficient neuron utilization, and rapid task-specific adaptation. 2) We propose population-based mask evolution for efficient task-specific connectivity optimization, and construct a mask-based policy population for efficient policy learning. Furthermore, Nevo-CRL achieves high neuron efficiency by pruning low-importance connections to recycle neurons. 3) We propose a semantic similarity-based mask construction method to enable efficient knowledge reuse across tasks. To prevent forgetting, previously learned neurons and their connections are frozen. 4) We empirically demonstrate that Nevo-CRL outperforms existing methods across different metrics.

## 2. Background

### 2.1. Preliminaries

We follow the task-incremental setting adopted by prior work (Khetarpal et al., 2022; Wolczyk et al., 2022b; Wołczyk et al., 2021; Rolnick et al., 2019; Mendez et al., 2020; Serra et al., 2018; Schwarz et al., 2018). Consider a sequence of $N$ tasks $\{\mathcal{T}_1, \cdots, \mathcal{T}_N\}$, where each task $\mathcal{T}_i$ is formulated as a Markov Decision Process (MDP) defined by a tuple $\mathcal{T}_i = \langle \mathcal{S}_i, \mathcal{A}_i, \mathcal{P}_i, \mathcal{R}_i, \gamma, T \rangle$. In each step $t$ of task $\mathcal{T}_i$, the agent selects an action $a_{i,t} \sim \pi_\theta(\cdot \mid s_{i,t}) \in \mathcal{A}_i$ according to the current state $s_{i,t} \in \mathcal{S}_i$, then the environment transitions to $s_{i,t+1} \sim \mathcal{P}_i(s_{i,t}, a_{i,t})$, and the agent receives a reward $r_{i,t} = \mathcal{R}_i(s_{i,t}, a_{i,t})$. $T$ denotes the maximum episode horizon. The objective of continual RL is to learn a policy $\pi_\theta$ at task $\mathcal{T}_N$ that achieves high expected returns on all previously encountered tasks ($i \leq N$), under the constraint of having limited or no access to past experiences. The optimization objective can be defined as $\theta^* = \arg\max_\theta \sum_{i=1}^N \mathbb{E}_{\pi_\theta}\left[\sum_{t=1}^T \gamma^t \mathcal{R}_i(s_{i,t}, a_{i,t})\right]$. Despite notable successes, existing RL methods remain limited in two key aspects: rapid generalization to novel tasks using prior knowledge, and resistance to catastrophic forgetting during continual learning.

### 2.2. Related Works

Existing Continual Learning methods can be broadly categorized into three categories: rehearsal-based methods, regularization-based methods, and structure-based methods.

**Rehearsal-based methods** mitigate forgetting in continual learning by storing experiences from previous tasks and replaying them during training. CLEAR (Rolnick et al., 2019) achieves efficient replay by integrating on-policy

learning, off-policy learning, and behavior cloning. A-GEM (Chaudhry et al., 2019) constrains policy gradient updates by projecting them onto the closest direction that guarantees no decrease in average performance on past tasks. ClonEx-SAC (Wolczyk et al., 2022a) alleviates forgetting by retaining samples from earlier tasks and applying behavior cloning. These methods typically require higher memory, which limits their applicability in some scenarios.

**Regularization-based methods** mitigate forgetting by adding penalty terms to the loss function. These penalties discourage significant changes to parameters important for previous tasks, thus reducing policy drift. EWC (Kirkpatrick et al., 2016) uses the Fisher information matrix to identify and selectively constrain important parameters from previous tasks, rather than penalizing all parameters equally. MAS (Aljundi et al., 2018) dynamically applies parameter-wise regularization, where the importance of each parameter is estimated based on its effect on the policy's output. VCL (Nguyen et al., 2018) employs variational inference to constrain updates through KL divergence between current and past parameter distributions.

**Structure-based methods** contain Weight-level & Neuron-level methods. Weight-level methods share a common backbone network and use task-specific binary weight masks. PackNet (Mallya & Lazebnik, 2018) prunes and freezes important weights after each task to free capacity for future ones. SupSup (Wortsman et al., 2020) keeps the backbone fixed and learns only a supermask for every new task. WSN (Kang et al., 2022) trains weights and masks jointly, then compresses the masks with Huffman coding so memory grows sub-linearly with the number of tasks. Neuron-level methods mask the layer outputs rather than policy weights, yielding more compact task sub-networks. PathNet (Fernando et al., 2017) finds a neuron path for each task via evolutionary search and then freezes it for reuse. HAT (Serra et al., 2018) learns hard-attention masks jointly with weights through gradient descent to block interference. CTR (Ke et al., 2021) inserts adapter-style modules and masks to enable parameter-efficient transfer while avoiding forgetting. SpaceNet (Sokar et al., 2021) trains sparse neuron subsets per task and compresses the resulting masks so memory grows sub-linearly. CoTASP (Yang et al., 2023) employs dictionary learning to construct the sparse mask for each layer. CoTASP achieves state-of-the-art performance on the Continual World benchmark.

Nevo-CRL belongs to neuron-level methods and simulates signal transmission in the brain by applying masks to the layer output. Nevo-CRL differs from the above methods in several key aspects. First, Nevo-CRL constructs an evolutionary mask population through a semantic-similarity-based mask construction method, and proposes importance-guided crossover for iterative optimization. In addition, Nevo-CRL integrates pruning directly into the RL process and selectively reuses prior parameters, rather than reactivating all previously learned parameters. Finally, Nevo-CRL introduces an experience-sharing mechanism to further improve sample efficiency.

**Evolutionary Reinforcement Learning** (ERL) refers to a family of hybrid algorithms that integrate evolutionary algorithms (EAs) with RL. It constitutes a broad research area encompassing multiple directions (Li et al., 2024a), including EA-assisted RL, RL-assisted EA, and synergistic optimization of EAs and RL. For example, representative studies have explored efficient policy search (Khadka & Tumer, 2018; Pourchot & Sigaud, 2019; Bodnar et al., 2020; Hao et al., 2023; Li et al., 2023; 2024b;c), reward function design (Niekum et al., 2010; Ma et al., 2024; Li et al., 2025; 2026b), and other optimization problems (Zheng et al., 2019; Li et al., 2026a). Nevo-CRL also falls under the collaborative optimization branch of ERL. It decouples the continual learning problem into two parts: EAs optimize the masks to improve connectivity, while RL learns the policy parameters. To the best of our knowledge, Nevo-CRL is the first ERL framework specifically designed for continual problems.

# 3. Continual Evolutionary Reinforcement Learning

This section provides an overview of Nevo-CRL. We first introduce the optimization process of Nevo-CRL. Then, we provide a detailed description of several key components of Nevo-CRL, including sparse mask construction, population-based mask evolution and policy learning. Finally, we present the algorithm pseudocode.

### 3.1. Nevo-CRL Framework

Inspired by neuroscience, we propose Nevo-CRL, which maintains a fixed-capacity monolithic policy and assigns task-specific sparse masks to each new task. A mask is a binary indicator over network neurons, consisting of activated and inactive units. A value of 1 activates the corresponding neuron, while 0 does not. The masks operate on the outputs of each neural network layer, mimicking the brain's neural circuits. By jointly optimizing the sparse masks and the activated neurons, Nevo-CRL enables efficient learning of new tasks. The overall architecture is illustrated in Figure 1. Specifically, Nevo-CRL consists of four key stages:

**Sparse Mask Construction**. Based on the task description, we generate a semantic representation using a pretrained language model. We match the learned masks from previous tasks based on representation similarity, thereby reactivating well-trained neurons. These reused neurons are frozen to prevent forgetting. To preserve plasticity, we activate a subset of unused neurons, and integrate the corresponding

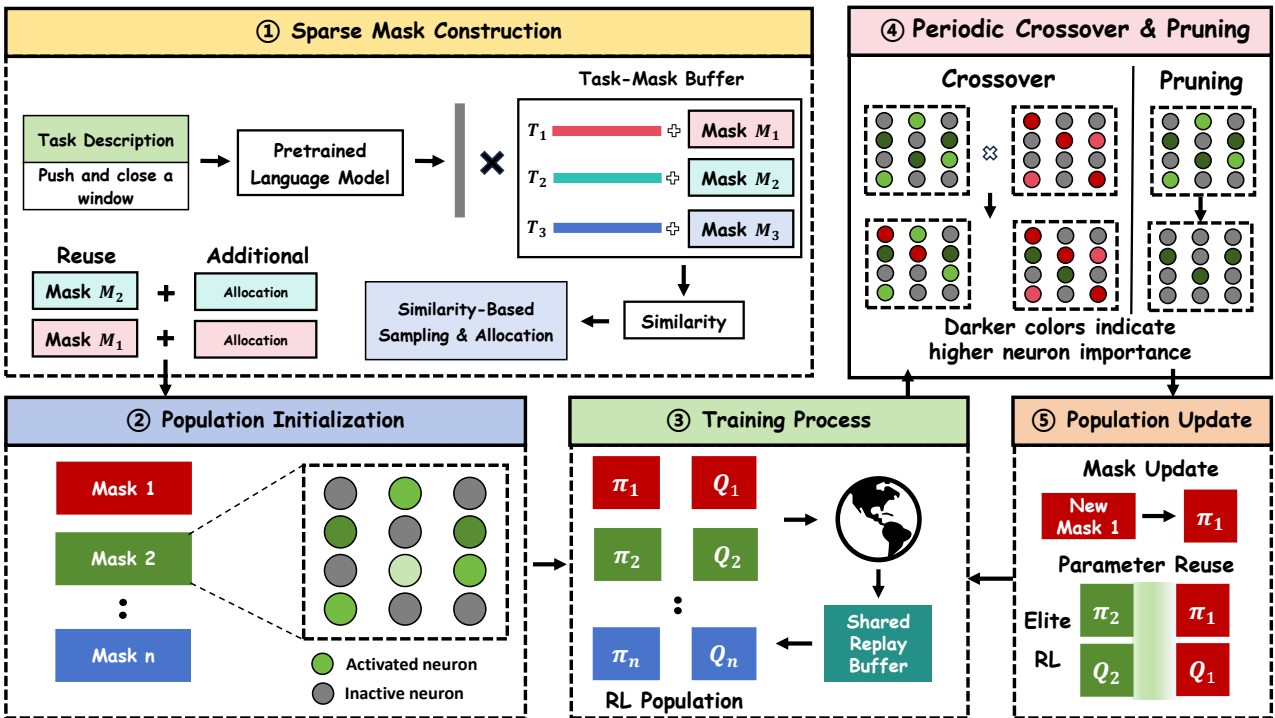

*Figure 1.* The overall framework of Nevo-CRL consists of five main steps: ① Reuse masks based on task semantic similarity and allocate additional parameters to construct a sparse mask population. ② Initialize the RL agent population based on the monolithic policy and apply the corresponding masks for subsequent learning. ③ The RL population interacts with the environment, and the collected experiences are stored in a shared replay buffer. ④ Periodically select elite masks, perform importance-guided crossover, and prune unimportant elements in all masks to enhance sparsity. ⑤ Update the generated masks to non-elite individuals and reuse parent parameters. Steps ③, ④, and ⑤ are iterated continuously. When a new task arrives, the process restarts from ①.

masks with the selected learned mask to construct the mask for the current task.

**Population Initialization**. To enable effective mask optimization and efficient policy learning, Nevo-CRL constructs a mask population $\mathbb{P}_{\text{mask}}$. We replicate the monolithic policy and pair it with different masks to construct task-specific policies. Each policy is paired with a distinct critic and trained using RL, i.e., SAC. Through this initialization, we create an RL population $\mathbb{P}_{\text{RL}}$.

**Mask Evolution & Pruning**. During training, masks are periodically optimized via crossover among elite masks every $T_{\text{Interval}}$ environment steps to improve task-specific connectivity. Additionally, we prune low-importance activated units within the masks to recycle neurons, thereby enhancing neuron efficiency.

**Population Update and Optimization**. The neural connections of underperforming individuals are updated using the crossover-generated masks. To improve learning efficiency, we merge parameters from elite policies into these individuals, avoiding learning from scratch.

Through these stages, Nevo-CRL enables rapid adaptation to new tasks while preserving performance on previously learned policies. In the following sections, we detail two core components of Nevo-CRL: Sparse Mask Construction and Mask Optimization & Policy Learning.

### 3.2. Sparse Mask Construction

Motivated by the brain's sparse connectivity and neuron reuse, we construct a sparse mask that is applied to the outputs of each network layer, mimicking brain-like neural circuits. Specifically, given the monolithic policy network with $L$ hidden layers, we denote $\mathbf{y}^{(l)}$ and $\boldsymbol{\theta}^{(l)}$ as the output vector and weights of layer $l$. For each task $i$, a binary mask $m_i^{(l)}$ is generated for every layer and applied element-wise to the layer output:

$$\mathbf{y}^{(l+1)} = f\big(\boldsymbol{m}_i^{(l)} \odot \mathbf{y}^{(l)}; \boldsymbol{\theta}^{(l)}\big), \tag{1}$$

where $f$ represents the standard layer operation and $\odot$ denotes element-wise multiplication. The connectivity and activated neurons across all layers define a task-specific policy for task $i$.

To enable efficient reuse of previously learned neurons, we propose a task-semantic-similarity-based mask construction method. For each task $\mathcal{T}_i$, we first use a pretrained Sentence-BERT (Reimers & Gurevych, 2019) to encode

the task description into a semantic representation $e_i$. All previous task representations and their associated masks are stored in a task-mask archive $\mathcal{A}$ for future retrieval. Notably, semantic representations are not used as inputs to the policy. When a new task $\mathcal{T}_j$ arrives, we first obtain its semantic representation $e_j$. We then compute the cosine similarity between $e_j$ and each task representation stored in the archive $\mathcal{A}$. We select tasks with high semantic similarity $s_i$ and reuse their corresponding mask $m_i$. To mitigate forgetting, neurons activated by previously used masks are frozen. For a new task $\mathcal{T}_j$, we randomly select $n_j^{(l)}$ units in each hidden layer $l$ to ensure policy plasticity. These units are merged with the matched learned mask $m_i$ to form the final mask $m_j$ for $\mathcal{T}_j$. The number $n_j^{(l)}$ is computed adaptively from the task-similarity score $s_i$:

$$n_j^l = n_{\text{dim}}^l \cdot (p_{\min} + (p_{\max} - p_{\min}) \cdot (1 - s_i)), \quad (2)$$

where $n_{\text{dim}}^l$ is the output dimensionality of layer $l$, and $p_{\min}$ and $p_{\max}$ are user-defined lower and upper allocation ratios with $0 < p_{\min} \leq p_{\max} \leq 1$. The similarity $s_i$ is clipped at 0.0. Based on this formula, fewer neurons are allocated to similar tasks to promote reuse, while dissimilar tasks receive more to support new skill learning. Through the procedure, we obtain the initial mask $m_j$ for task $\mathcal{T}_j$.

### 3.3. Mask Evolution and Policy Learning

During learning, the human brain continuously eliminates existing synapses, forms new ones to construct new neural circuits, and dynamically adjusts the strength of neuronal signaling. Inspired by this, continual optimization of both masks and neurons—adapting connectivity and signal transmission—is essential for supporting lifelong learning.

**Population Initialization**. To this end, we first propose the population-based mask evolution method for optimizing connectivity. We first initialize the mask population $\mathbb{P}_{\text{mask}}$. Specifically, we compute the similarity between representations and apply a softmax to obtain sampling probabilities over the learned tasks. We then sample $n$ task masks accordingly and augment each with additional inactive units, forming the initial mask population $\mathbb{P}_{\text{mask}} = \{m_1, \cdots, m_n\}$ for task $\mathcal{T}_j$. In the early stages of training, the number of previously learned tasks may be smaller than the population size. To maintain a consistent population structure, we perform sampling with replacement. For the first task, the population is initialized by randomly sampling $n$ masks. Next, we replicate the monolithic policy $n$ times and combine each copy with a corresponding mask from $\mathbb{P}_{\text{mask}}$ to construct task-specific policy population $\mathbb{P}_{\text{RL}}$. Each policy maintains an individual critic for value function approximation. The policy population is used for parameter optimization, while the mask population focuses on optimizing the connectivity circuit of each policy.

**Parameter Learning**. Each policy interacts with the environment, and the collected experiences are stored in a shared replay buffer $\mathcal{D}$. All individuals then learn from the data sampled from $\mathcal{D}$. This approach improves the experience diversity and sample efficiency.

**Mask Evolution**. We record the performance $\{f_1, \cdots, f_n\}$ (e.g., success rate) of each policy as the fitness to evaluate $\mathbb{P}_{\text{mask}}$. Every $T_{\text{Interval}}$ environment steps, we perform mask evolution to optimize connectivity. To improve efficiency, we propose the importance-guided crossover method. Specifically, we first select the top two masks, $m_{p_1}$ and $m_{p_2}$, based on the fitness. Next, we sample a large batch of experiences (e.g., 10,240) from the replay buffer $\mathcal{D}$. For each active unit in the mask, we compute the average absolute values $v_{p_1}$ and $v_{p_2}$ through the forward propagation of the policy network. A low value indicates that the connection contributes minimally to the output and is gradually becoming inactive, while a high value indicates that the unit has a significant influence on the final output.

After the above preparation, we first remove the shared active units $m_c$ from the parent masks $m_{p_1}$ and $m_{p_2}$, yielding their exclusive activation masks $m_{u_1}$ and $m_{u_2}$. Based on the corresponding importance scores $v_{p_1}$ and $v_{p_2}$, we select the top $k\%$ most important units from $m_{u_1}$ and $m_{u_2}$, denoted as $m_{\text{top}_1}$ and $m_{\text{top}_2}$. The remaining exclusive units are denoted as $m_{\text{no}_1} = m_{u_1} \setminus m_{\text{top}_1}$ and $m_{\text{no}_2} = m_{u_2} \setminus m_{\text{top}_2}$. We then perform random crossover between $m_{\text{top}_1}$ and $m_{\text{top}_2}$ to generate two offspring components, $m'_{c_1}$ and $m'_{c_2}$. Finally, the offspring masks $m_{c_1}$ and $m_{c_2}$ are constructed as:

$$m_{c_1} = m_c \cup m_{\text{no}_1} \cup m'_{c_1}, \quad m_{c_2} = m_c \cup m_{\text{no}_2} \cup m'_{c_2}$$

This approach facilitates mask optimization within a substantially compressed search space, thereby enhancing the efficiency of identifying task-optimal connectivity.

**Population Update**. After the crossover, the offspring masks are assigned to non-elite individuals in $\mathbb{P}_{\text{RL}}$ for optimizing connectivity. To avoid training policies from scratch, we fuse the policy parameters of the parent policies. Specifically, for each activated unit in an offspring mask, we identify whether it originates from the first or second parent mask, and copy the corresponding parameters from the matched parent policy. The offspring also inherits the critic network directly from the first parent. Through this process, we achieve both connectivity optimization and neuron adjustment, enabling efficient optimization.

**Mask Pruning**. During learning, we perform pruning on all masks in the population. Specifically, any unit whose activation level falls below a threshold $b$ is switched from 1 to 0. This pruning mechanism removes low-activity connections and further reduces the number of active neurons, which improves parameter efficiency and enables the model to accommodate more tasks within the same network capacity.

## 3.4. Algorithm Pseudocode

To provide a clear procedure of Nevo-CRL, we present its pseudocode in Algorithm 1. Specifically, the process consists of the following key stages: For each task, we first generate a mask population $\mathbb{P}_{\text{mask}}$ based on the proposed similarity-based mask construction method, and then construct a task-specific RL population $\mathbb{P}_{\text{RL}}$. Next, we proceed to the learning process. Each policy in $\mathbb{P}_{\text{RL}}$ interacts with the environment, and the collected experiences are stored in the shared replay buffer $\mathcal{D}$. All policies are then updated using data sampled from $\mathcal{D}$. The performance of each policy is used as the fitness for $\mathbb{P}_{\text{mask}}$. The top two masks are selected, and importance-guided crossover is applied to generate new offspring masks. Non-elite individuals are updated using the offspring masks and fuse policy parameters from the corresponding parents. The critic networks are also replaced. Finally, all masks undergo pruning, removing low-activation connections and recycling the corresponding neurons. The above procedure outlines the complete optimization cycle of Nevo-CRL. In the next section, we present a comprehensive experimental evaluation to validate its effectiveness.

## 4. Experiments

### 4.1. Experiment Setup

**Benchmarks**. We follow the experimental benchmark from previous works (Wolczyk et al., 2022a; Yang et al., 2023) and use Continual World (CW) (Wołczyk et al., 2021) for comprehensive experimental evaluation and analysis. We choose representative tasks CW10 and CW20 for evaluation. CW10 includes 10 diverse tasks from Metaworld, while CW20 is a repetition of CW10 to measure the transferability of the policy when encountering the same tasks. We follow the CoTASP (Yang et al., 2023) setup and use the pre-configured transfer, which shows a high variation of forward transfer both globally and locally throughout the sequence. Each task is trained with up to 1 million environment interaction steps. In addition, we evaluate Nevo-CRL on 12 task sequences from Brax (Gaya et al., 2023). Similar to CW, Brax assesses four aspects of continual learning: forgetting, transfer, robustness, and compositionality, testing whether agents can retain previous skills, avoid negative transfer, remain stable under distractions, and reuse prior skills for more complex tasks.

**Baselines**: For CW, we evaluate Nevo-CRL against several state-of-the-art (SOTA) baselines, categorized into three groups: 1) Regularization-based methods: L2 (Kirkpatrick et al., 2016), EWC (Kirkpatrick et al., 2017), MAS (Aljundi et al., 2018), VCL (Nguyen et al., 2018). 2) Structure-based methods: PackNet (Mallya & Lazebnik, 2018), Co-TASP (Yang et al., 2023). 3) Rehearsal-based methods: Reservoir (Wołczyk et al., 2021), A-GEM (Chaudhry et al.,

---

**Algorithm 1** Nevo-CRL Framework

1: **Require**: The monolithic policy $\pi$, Task–Mask Archive $\mathcal{A}$, language model M, replay buffer $\mathcal{D}$
2: **Hyperparameters**: Population size $n$, crossover interval $T_{\text{Interval}}$, pruning step $T_{\text{Pruning}}$, ratio $k\%$, threshold $b$
3: **for each** task $\mathcal{T}_j \in \{\mathcal{T}_1, \dots, \mathcal{T}_N\}$ **do**
4:     // Mask & policy population construction
5:     Initialize $\mathbb{P}_{\text{mask}}$ with similarity-based construction
6:     Initialize $\mathbb{P}_{\text{RL}}$ based on $\mathbb{P}_{\text{mask}}$ and $\pi$
7:     // Mask Evolution and Policy Learning
8:     **for** $t \leftarrow 1$ **to** $T$ **do**
9:         Policies in $\mathbb{P}_{\text{RL}}$ interact with the environment
10:         Store transitions in $\mathcal{D}$ and update actors/critics
11:         // importance-guided Crossover
12:         **if** $t \bmod T_{\text{Interval}} = 0$ **then**
13:             Select best two masks $m_{p_1}$ & $m_{p_2}$
14:             Apply importance-guided crossover
15:             Replace non-elite masks with child masks
16:             Update parameters of non-elite RL
17:         //Pruning to remove low-activity connections
18:         **if** $t = T_{\text{Pruning}}$ **then**
19:             Prune connections in $\mathbb{P}_{\text{mask}}$ (activation $< b$)
20:             Update the masks in $\mathbb{P}_{\text{RL}}$

---

2019), ClonEx-SAC (Wolczyk et al., 2022a). For thoroughness, we also include a simple sequential training approach (i.e., Fine-tuning) and key multi-task RL baselines MTL (Yu et al., 2020) and MTL+PopArt (Hessel et al., 2019), which are generally regarded as the soft upper bound for continual RL methods. For Brax tasks, we mainly compare Nevo-CRL with CoTASP, the best-performing structure-based continual RL method. For a fair comparison, all algorithms use SAC (Yarats & Kostrikov, 2020) as the RL backbone and run with five random seeds. For training settings and network architecture, we follow CoTASP without any changes.

**Metrics**. We follow the widely-used evaluation protocol in continual learning literature (Wołczyk et al., 2021) and use three metrics: 1) **Average Performance**: The average performance at time $t$ is defined as $\text{P}(t) := \frac{1}{N} \sum_{i=1}^{N} p_i(t)$, where $p_i(t) \in [0, 1]$ represents the success rate of task $i$. We report the average of the best performance achieved in each run. 2) **Forgetting**: This metric measures the average performance drop after each task has been learned, defined as $F = \frac{1}{N} \sum_{i=1}^{N} (p_i(i \cdot \delta) - p_i(\mathcal{T}_i \cdot \delta))$. 3) **Generalization**: Generalization refers to the average number of steps required to reach the success threshold across all tasks. Note that training is terminated once the success rate reaches the threshold (set to 0.8) in two consecutive evaluations. For more implementation details, architecture, and hyperparameters, please refer to Appendix B and C.

*Table 1.* Benchmark evaluation results on Continual World.

| Benchmarks | | CW 10 | | | CW 20 | | |
|---|---|---|---|---|---|---|---|
| **Metrics** | | $P\,(\uparrow)$ | $F\,(\downarrow)$ | $G\,(\downarrow)$ | $P\,(\uparrow)$ | $F\,(\downarrow)$ | $G\,(\downarrow)$ |
| **Reg** | L2 | 0.47±0.13 | 0.02±0.06 | 0.69±0.09 | 0.43±0.11 | 0.02±0.04 | 0.73±0.08 |
| | EWC | 0.64±0.12 | 0.05±0.11 | 0.45±0.06 | 0.60±0.10 | 0.02±0.07 | 0.55±0.06 |
| | MAS | 0.56±0.08 | 0.00±0.06 | 0.59±0.06 | 0.51±0.06 | 0.00±0.04 | 0.67±0.03 |
| | VCL | 0.51±0.11 | 0.01±0.05 | 0.62±0.09 | 0.48±0.06 | 0.01±0.04 | 0.65±0.06 |
| | FT | 0.12±0.01 | 0.83±0.04 | 0.34±0.05 | 0.05±0.00 | 0.73±0.04 | 0.42±0.04 |
| **Struc** | PackNet | 0.83±0.06 | 0.00±0.03 | 0.38±0.05 | 0.80±0.04 | 0.00±0.02 | 0.41±0.04 |
| | CoTASP | 0.89±0.05 | 0.00±0.00 | 0.47±0.01 | 0.86±0.03 | 0.00±0.00 | 0.43±0.04 |
| **Reh** | Reservoir | 0.17±0.10 | 0.01±0.05 | 0.83±0.05 | 0.12±0.08 | 0.07±0.06 | 0.89±0.03 |
| | A-GEM | 0.14±0.02 | 0.84±0.04 | 0.33±0.04 | 0.07±0.02 | 0.75±0.06 | 0.43±0.04 |
| | ClonEx-SAC | 0.86 | 0.02 | — | 0.87 | 0.02 | — |
| **MT** | MTL | 0.52±0.10 | — | — | 0.50±0.11 | — | — |
| | MTL+PopArt | 0.70±0.14 | — | — | 0.66±0.17 | — | — |
| **Nevo-CRL** | | **0.96**±0.01 | 0.00±0.00 | 0.41±0.01 | **0.93**±0.00 | **-0.02**±0.01 | **0.38**±0.02 |

*Table 2.* Performance comparison between CoTASP and Nevo-CRL on 12 Brax task sequences.

| Method | Ant | | | | | | Humanoid | | HalfCheetah | | | |
|---|---|---|---|---|---|---|---|---|---|---|---|---|
| | Forg. | Trans. | Robust. | Comp. | Dras.-2 | Dras.-1 | Official | Diff. | Forg. | Robust. | Trans. | Comp. |
| CoTASP | 2746 | 1315 | 2446 | 981 | 1615 | 2331 | 4192 | 3385 | 1679.11 | 4043.43 | 780.71 | 3936.48 |
| Nevo-CRL | **4003** | **3025** | **4728** | **2200** | **3871** | **4134** | **4826** | **3760** | **3341.40** | **5671.68** | **3211.93** | **4277.75** |

## 4.2. Performance Evaluation

In this section, we compare Nevo-CRL with different categories of SOTA baselines. Our primary focus is on performance, forgetting, and generalization ability. Table 1 summarizes the experimental results of different algorithms, where we observe that Nevo-CRL significantly outperforms other baselines across tasks with different sequence lengths. Besides, Nevo-CRL does not exhibit any forgetting problems. Compared to structure-based methods with the same network structure, Nevo-CRL demonstrates a substantial advantage in both final performance and generalization metrics. Nevo-CRL is the only algorithm capable of achieving over 90% performance on CW20. Additionally, as shown in Figure 2, Nevo-CRL achieves performance comparable to that of multi-task learning baselines at 10 million interaction steps, highlighting its efficient knowledge utilization.

Furthermore, we provide a detailed comparison of Nevo-CRL and CoTASP (the current best structure-based CRL method) for each task during learning. The results are shown in Figure 3. The red and blue shaded areas indicate positive transfer, while the green shaded area represents negative transfer. We observe that, except for one task, Nevo-CRL generally exhibits a stronger positive transfer advantage across tasks. In contrast, CoTASP demonstrates negative

transfer on more tasks and shows a weaker advantage in terms of positive transfer. Overall, Nevo-CRL achieves more efficient knowledge reuse than both CoTASP and single-task learning methods.

Besides, we provide the curve of the used capacity ratio versus performance in Figure 5. We observe that Nevo-CRL achieves better results with fewer parameters compared to other structure-based methods, demonstrating that Nevo-CRL is more parameter-efficient, capable of learning more skills with the same capacity.

For evaluation on Brax, we consider the HalfCheetah, Ant, and Humanoid domains, which are designed to assess different aspects of continual learning capability. The detailed task sequences are provided in Appendix E. The experimental results, reported as average returns over 5 seeds, are summarized in Table 2. Across all sequence types in the three domains, Nevo-CRL consistently outperforms CoTASP, the strong structure-based continual RL baseline.

## 4.3. Ablation & Analysis

Nevo-CRL consists of the following four key components: semantic similarity-based sparse mask construction, population-based optimization and learning, importance-

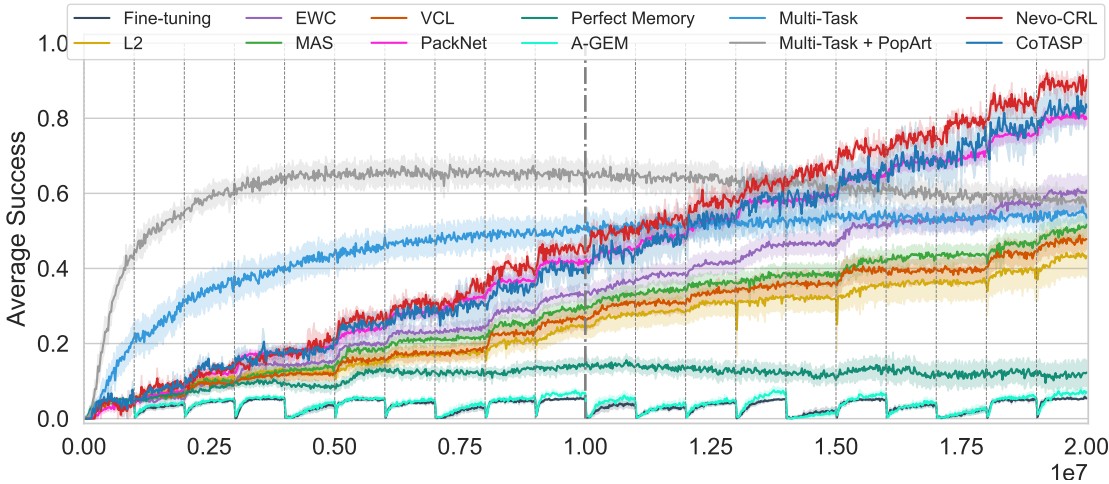

*Figure 2.* Performance comparison between Nevo-CRL and other baselines.

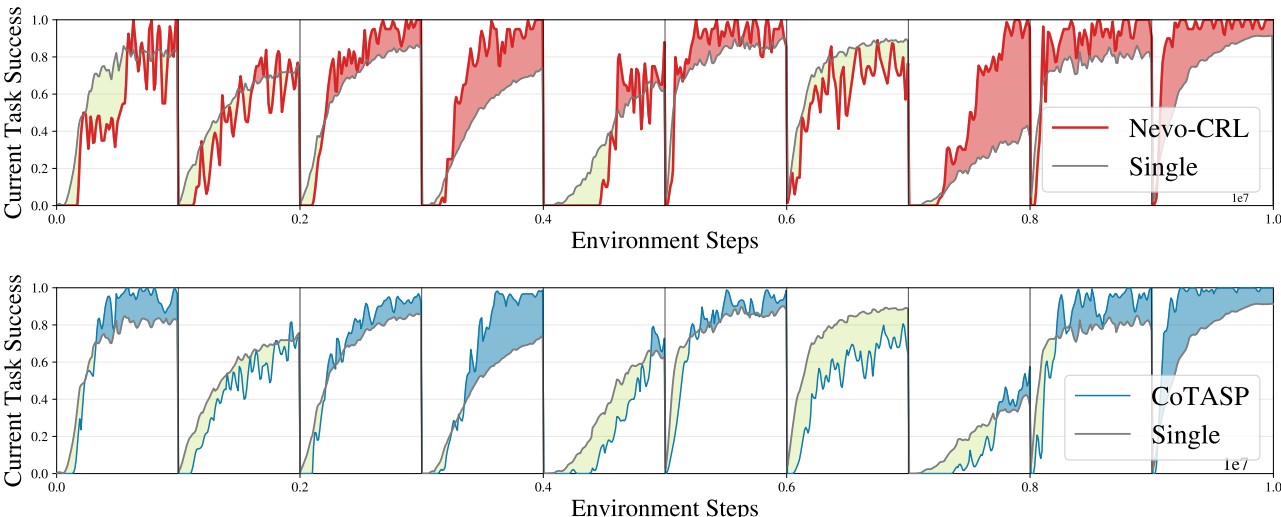

*Figure 3.* Comparison of training curves for Nevo-CRL and CoTASP against single-task learning across different tasks. Nevo-CRL exhibits more efficient positive transfer compared to CoTASP.

guided crossover, and pruning. We first conduct ablation studies on Nevo-CRL to answer the following questions: Q1: Is the mask construction based on semantic similarity efficient? Q2: Is the population-based optimization and learning in Nevo-CRL necessary? Q3: Is the importance-guided crossover more efficient than standard mask crossover methods? Q4: Is pruning necessary in Nevo-CRL?

To answer Q1, we remove our mask construction method and replace it with random mask construction, which randomly reuses the learned masks and allocates a fixed number of additional parameters (i.e., $p_{\max} \cdot n_{dim}^l$). As illustrated in Figure 4a, removing our mask construction leads to a clear performance drop, mainly due to negative transfer caused by the random reuse of unrelated knowledge. To answer

Q2, we remove the population in Nevo-CRL and retain only a single mask and a single RL individual. As shown in Figure 4b, the performance drops significantly without the population. The main reason is that removing the population removes both mask evolution and efficient policy exploration, which leads to a decline in performance. This demonstrates the necessity of population-based mask evolution and parameter optimization. To answer Q3, we replace the importance-guided crossover with random crossover. The results in Figure 4c show that random crossover results in a notable performance drop. This is due to the larger and more complex search space introduced by random crossover, which makes optimization more difficult. In contrast, importance-guided crossover restricts the search to more promising regions, making it easier to discover high-

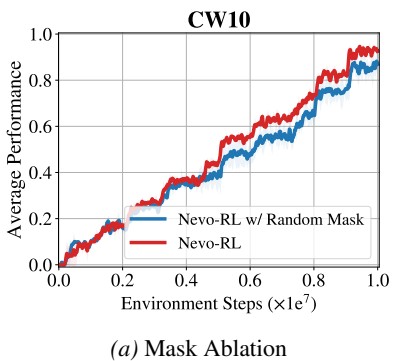
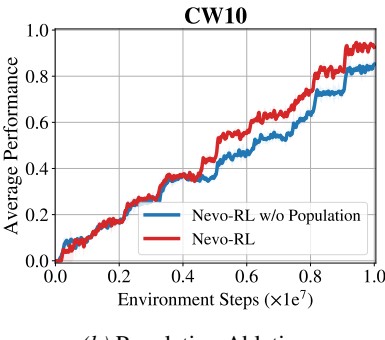
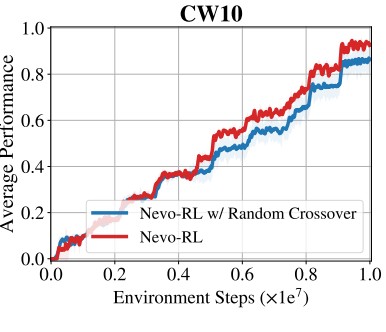

*(a)* Mask Ablation  *(b)* Population Ablation  *(c)* Crossover Ablation

*Figure 4.* Ablation study on Nevo-CRL.

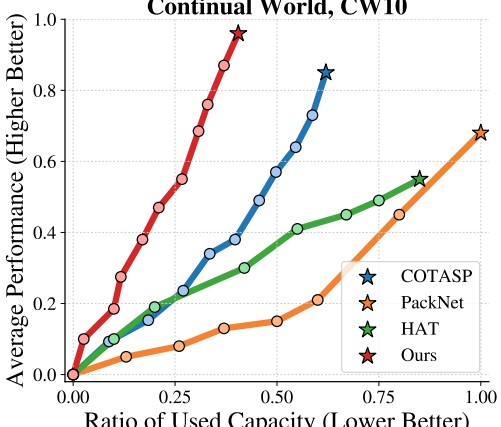

*Figure 5.* Comparison of Performance under Varying Parameter Utilization.

quality solutions. To answer Q4, we examine the effect of removing pruning. Without pruning, the network quickly reaches full memory utilization (100%) after only a few tasks. While incorporating pruning may cause temporary performance fluctuations, the overall performance remains comparable to that of the non-pruned variant. Therefore, pruning is necessary to ensure sustainable memory usage and maintain long-term learning capacity.

### 4.4. Computational Overhead

Nevo-CRL is implemented in JAX and uses vmap to accelerate population updates via multi-policy parallelism. Although this design improves efficiency, it still introduces additional computational overhead. Specifically, CoTASP requires 45.9 hours to complete CW20, whereas Nevo-CRL requires 70.8 hours, representing a 54% increase in training time. Regarding memory overhead, the shared replay buffer prevents additional main memory consumption. In terms of GPU memory, the population training leads to a usage of approximately 2.6 GB on a single NVIDIA 4090 GPU.

## 5. Conclusion

We propose a novel neuro-evolutionary continual RL framework Nevo-CRL that integrates neuroscience principles to tackle catastrophic forgetting and enable rapid task adaptation. To promote effective knowledge reuse, we propose semantic similarity-based mask construction that selectively activates previously learned neurons. Nevo-CRL constructs a mask population and proposes importance-guided crossover to optimize neural connectivity. Besides, Nevo-CRL constructs an RL population based on the masks to enable parameter optimization. To improve efficiency, Nevo-CRL performs pruning to eliminate low-activity connections and recycle unused neurons. Experimental results demonstrate that Nevo-CRL outperforms strong baselines, establishing SOTA performance.

## Impact Statement

This paper presents work whose goal is to advance the field of Machine Learning. There are many potential societal consequences of our work, none which we feel must be specifically highlighted here.

## Acknowledgments

This work is supported by the National Natural Science Foundation of China (Grant Nos. 624B2101, 62422605, 62533021, 62541610), the National Key Research and Development Program of China (Grant No. 2024YFE0210900) and the MoE Key Laboratory of Brain-inspired Intelligent Perception and Cognition, University of Science and Technology of China (No. 2521006). We would like to thank all the anonymous reviewers for their valuable comments and constructive suggestions, which have greatly improved the quality of this paper.

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

## A. Limitations & Future Work

While our work offers promising empirical results, it has several limitations that warrant future research. First, the current framework is largely heuristic in nature and still lacks rigorous theoretical justification and formal analysis. Second, the neuroscience-inspired designs adopted in this paper represent only one possible instantiation; further analysis and optimization of these components are still needed. Third, Nevo-CRL maintains a population of policies, which inevitably introduces additional computational overhead. Although parallel training in JAX helps reduce this cost, the overall training time is still about 50% higher than that of non-population methods. Fourth, Nevo-CRL inherits a common limitation of structure-based approaches: it currently lacks an effective mechanism for handling cases where network capacity is fully utilized. How to expand the network without affecting previously learned policies, or alternatively compress existing policies through parameter fusion, remains an important direction for future work. Finally, the scope of this study is limited to collaborative robotic tasks due to their application potential, and broader evaluation across diverse domains is still needed. We leave these aspects as important directions for future work.

## B. Training & Evaluation Details

**Implementation Details of Nevo-CRL**: Nevo-CRL is implemented in JAX [1] and is built upon the official codebase of CoTASP [2]. All hyperparameter settings and network architectures strictly follow the original implementation without any modifications. The Nevo-CRL-specific hyperparameters are introduced in Section 4.

To ensure fairness, all algorithms are evaluated using the same SAC framework. The actor and the critic are implemented as two separate multi-layer perceptron (MLP) networks, each consisting of 4 hidden layers with 256 neurons. For structure-based methods (PackNet, CoTASP) and our proposed Nevo-CRL, a wider MLP with 1024 neurons per layer is used for the actor. These configurations strictly follow prior work and are kept unchanged.

The detailed task list and descriptions are as follows:

*Table 3.* This table lists all Continual World tasks along with their descriptions. The sequence shown corresponds to CW10. The CW20 sequence consists of the same tasks repeated twice. Tasks are learned sequentially, with 1 million environment interaction steps allocated per task.

| Index | Task | Description |
|-------|------|-------------|
| 1 | hammer-v1 | Hammer a screw on the wall. |
| 2 | push-wall-v1 | Bypass a wall and push a puck to a goal. |
| 3 | faucet-close-v1 | Rotate the faucet clockwise. |
| 4 | push-back-v1 | Pull a puck to a goal. |
| 5 | stick-pull-v1 | Grasp a stick and pull a box with the stick. |
| 6 | handle-press-side-v1 | Press a handle down sideways. |
| 7 | push-v1 | Push the puck to a goal. |
| 8 | shelf-place-v1 | Pick and place a puck onto a shelf. |
| 9 | window-close-v1 | Push and close a window. |
| 10 | peg-unplug-side-v1 | Unplug a peg sideways. |

During evaluation, the best-performing individual for the current task is not known a priori. Therefore, we evaluate the policy associated with each mask and report the highest performance across all masks. For previously learned tasks, we directly evaluate the best policy together with its corresponding mask.

## C. Hyperparameter Setting

This section lists the hyperparameters in Table 4. For SAC-related hyperparameters, we adopt the same configurations as those used in CoTASP. The specific settings for the Nevo-CRL-specific hyperparameters are also included. A detailed analysis of these hyperparameters is presented in Section D.

---

[1] https://github.com/jax-ml/jax

[2] https://github.com/stevenyangyj/cotasp

*Table 4.* Hyperparameters of Nevo-CRL for Continual World experiments

| Hyperparameter | Selected Value |
|---|---|
| *SAC Hyperparameters* | |
| Actor hidden size | 1024 |
| Critic hidden size | 256 |
| Layer size | 4 |
| Activation function | LeakyReLU |
| Batch size | 256 |
| Discount factor | 0.99 |
| Target entropy | $-2.0$ |
| Target interpolation | $5 \times 10^{-3}$ |
| Replay buffer size | $10^6$ |
| Exploratory steps | $10^4$ |
| Optimizer | Adam |
| Learning rate | $3 \times 10^{-4}$ |
| *Nevo-CRL-specific Hyperparameters* | |
| Top $k\%$ | 10 |
| Pruning threshold $b$ | 0.005 |
| Minimum selection ratio | 0.01 |
| Maximum selection ratio | 0.25 |
| Crossover interval | 200,000 |
| Population size | 5 |
| Pruning step | 700,000 |

*Table 5.* Analysis of the pruning step on CW20.

| Pruning step | 500,000 | 700,000 | 800,000 | 900,000 |
|---|---|---|---|---|
| Performance | 0.92 | 0.93 | 0.94 | 0.88 |
| Capacity Utilization Rate | 50% | 56% | 58% | 63% |

# D. Additional Experiments

In this section, we provide additional experiments to analyze several important hyperparameters. Specifically, we focus on the following six: pruning threshold $b$, pruning step $T_{\text{Pruning}}$, minimum selection ratio $p_{\min}$, maximum selection ratio $p_{\max}$, population size $n$, and crossover interval $T_{\text{Interval}}$. Among them, pruning threshold and pruning step are hyperparameters related to the pruning mechanism; minimum selection ratio and maximum selection ratio are hyperparameters used in mask construction; population size and crossover interval pertain to the population-based optimization component.

We begin by analyzing the pruning threshold and pruning step. It is important to emphasize that **the goal of the pruning mechanism is not to improve network plasticity, but rather to ensure more efficient utilization of neurons. Specifically, pruning is employed to remove less important neurons while maintaining overall performance. In this context, our objective is to eliminate redundancy without compromising the performance of the learned policy.** First, it is critical to avoid initiating pruning too early; the pruning step should be set to a relatively high value. We find that when pruning is triggered as early as 10,000 steps, nearly all parameters—close to 100%—are removed. This occurs because the network, still in its early training phase with randomly initialized parameters, has not yet acquired meaningful representations. Consequently, premature and aggressive pruning severely reduces the model's plasticity, significantly hindering its capacity for effective learning in later stages. We analyze the effect of different pruning step values, selecting from $\{500000, 700000, 800000, 900000\}$. The results are presented in Table 5. We find that moderately increasing the pruning step improves overall performance. This is because smaller pruning steps tend to eliminate more neurons prematurely, which reduces capacity utilization and ultimately degrades performance. However, we observe that an overly large pruning step results in performance deterioration, potentially due to the insufficient number of steps available for effective policy

*Table 6.* Analysis of the pruning threshold on CW20.

| Pruning Threshold | 0.001 | 0.0025 | 0.005 | 0.01 |
|---|---|---|---|---|
| Performance | 0.46 | 0.68 | 0.93 | 0.84 |
| Capacity Utilization Rate | 100% | 100% | 56% | 24% |

*Table 7.* Analysis of $p_{\max}$ and $p_{\min}$ on CW20.

| $p_{\max}$ | 0.1 | 0.25 | 0.3 |
|---|---|---|---|
| Performance | 0.89 | 0.93 | 0.89 |
| Capacity Utilization Rate | 57% | 56% | 60% |
| $p_{\min}$ | 0.01 | 0.05 | 0.1 |
| Performance | 0.93 | 0.86 | 0.89 |
| Capacity Utilization Rate | 56% | 72% | 75% |

adaptation after pruning. To balance the pruning effectiveness and network stability, we recommend setting the pruning step to 700,000 or 800,000. All experiments reported in this study are conducted with the pruning step set to 700,000, chosen for consistency, although other values (e.g., 800,000) may yield slightly better performance.

For the selection of the pruning threshold, the experimental results are shown in Table 6. A smaller bar value fails to enable effective pruning, ultimately preventing successful learning. For instance, when $b = 0.001$ or $0.0025$, the model reaches 100% capacity as early as tasks 9–10 (or tasks 13–16), leaving no remaining capacity for learning subsequent tasks. Conversely, setting the pruning threshold too high can also negatively affect performance, as it may lead to the removal of important neurons and a significant drop in capacity utilization. To balance the performance and utilization efficiency, we adopt a pruning threshold of 0.005 throughout all experiments.

Next, we evaluate the effect of the selection ratio in mask construction, focusing on $p_{\min}$ and $p_{\max}$. As shown in Table 7, both excessively small and large values of $p_{\max}$ can degrade performance. A smaller $p_{\max}$ leads to sparser connectivity and fewer active neurons, reducing the network's plasticity. In contrast, a larger $p_{\max}$ lowers sparsity, which may also hinder performance, as prior studies suggest that maintaining an appropriate level of sparsity benefits efficiency and performance. Moreover, changes in $p_{\min}$ have a greater impact on neuron allocation compared to $p_{\max}$. Increasing $p_{\min}$ results in higher capacity utilization and denser connectivity, but often at the cost of performance. In contrast, smaller values of $p_{\min}$ preserve sparsity and consistently yield better results. Based on this analysis, we set $p_{\max} = 0.25$ and $p_{\min} = 0.01$ throughout this study.

We further analyze the impact of several key hyperparameters in Nevo-CRL. First, we examine $k\%$ in the importance-guided crossover, which determines the proportion of top-ranked units selected. As shown in Table 9, we find that 10% outperforms both 5% and 20%. A larger $k$ increases the search space and makes optimization more difficult, while a too small $k$ risks falling into suboptimal solutions. Due to space limitations, analyses of other parameters are provided in Appendix D. Overall, for the pruning threshold $b$, a value that is too small fails to effectively prune, eventually causing the network to reach its maximum capacity and collapse, while a large value may directly remove important connections and lead to policy failure. For the mask construction parameter $p$, we recommend using a relatively large value to maintain sufficient network plasticity.

We then analyze the impact of population size and crossover interval, with the results summarized in Table 8. As observed, both small (population size = 2) and large (population size = 10) settings yield lower performance compared to a moderate population size of 5. This can be attributed to two main factors: larger population reduces the number of interaction steps allocated to each individual, limiting their learning efficiency; in contrast, smaller population lacks the diversity and capacity required for effective mask optimization.

Regarding the crossover interval, we compare two settings: 200,000 and 50,000. The larger interval of 200,000 allows sufficient time for the policy to adapt to the newly generated mask before the next crossover takes place. In contrast, the smaller interval of 50,000 results in overly frequent updates, which hinders proper adaptation and introduces instability in the learning process. Based on this analysis, we adopt a population size of 5 and a crossover interval of 200,000 throughout

*Table 8.* Analysis of population size and crossover interval on CW20.

| Setting | Performance | Capacity Utilization Rate |
|---|---|---|
| Population size = 2 | 0.88 | 58% |
| Population size = 5 | 0.93 | 56% |
| Population size = 10 | 0.90 | 61% |
| Crossover interval = 50,000 | 0.81 | 62% |
| Crossover interval = 200,000 | 0.93 | 56% |

*Table 9.* Analysis of top $k$ in importance-guided crossover on CW20.

| Hyperparameter | Value | Performance (CW20) |
|---|---|---|
| | 5 | 0.88 |
| Top $k$(%) in Crossover | 10 | 0.93 |
| | 20 | 0.90 |

this study.

Overall, maintaining the above hyperparameter settings consistently ensures superior performance across all 20 tasks in Continual World when compared to the baselines.

We further conduct ablation studies to isolate the effects of evolutionary mask search, neuron freezing, and neuron reuse. The results are shown in Table 10. Removing evolutionary mask search decreases the performance from 0.93 to 0.87, indicating that the evolutionary optimization process helps identify higher-quality masks. Removing neuron freezing leads to a severe performance drop, suggesting that the framework can no longer preserve previously learned tasks without freezing the corresponding neurons. Removing neuron reuse also substantially degrades performance and causes the capacity utilization rate to reach 100%, showing that reusing previously learned neurons is important for accommodating long task sequences under a fixed-capacity network.

## E. Brax Benchmark Details

Brax is inspired by the design philosophy of CW and constructs task sequences targeting different skill dimensions.

The following are some common task sequence scenarios in Brax:

- **Forgetting Scenarios** are designed such that a single policy tends to forget the former task when learning a new one.

- **Transfer Scenarios** are designed such that a single policy has more difficulty learning a new task after having learned the former one, rather than learning it from scratch.

- **Robustness Scenarios** alternate between a normal task and a very different distraction task that disturbs the whole learning process of a single policy (simply inverted the actions). In these tasks, fine-tuning policies struggle to recover good performance (the final average reward actually decreases).

- **Compositional Scenarios** present two first tasks that will be useful to learn the last one, but a very different distraction task is put at the third place to disturb this forward transfer. The last task is indeed a combination of the two first tasks in the sense that it combines their particularities.

Both HalfCheetah and Ant include the above scenarios.

The details of the Ant domain are as follows. Each sequence is repeated once, with 1 million steps per task.

- Forgetting: normal → hugefeet → rainfall → moon

- Transfer: nofeet_1_3 → nofeet_2_4 → nofeet_1_2 → nofeet_3_4

*Table 10.* Ablation study of evolutionary mask search, neuron freezing, and neuron reuse on CW20.

| CW20 | Nevo-CRL | w/o Evolutionary Mask Search | w/o Neuron Freezing | w/o Neuron Reuse |
|---|---|---|---|---|
| Performance | 0.93 | 0.87 | 0.14 | 0.45 |
| Capacity Utilization Rate | 56% | 56% | - | 100% |

- Robustness: normal → inverted_actions → normal → inverted_actions

- Compositional: nofeet_2_3_4 → nofeet_1_3_4 → nofeet_1_2 → nofeet_3_4

- Drastic change 1: hugefeet → moon → rainfall → nofeet_1_3

- Drastic change 2: nofeet_2_3_4 → moon → inverted_actions → hugefeet

The task sequences in the higher-dimensional Humanoid domain are as follows.

- Official version: normal → moon → carrystuff → tinyfeet (No repetition with 2 million steps per task)

- Different seq: tinyfeet → moon → carrystuff → normal (Each sequence is repeated once, with 1 million steps per task)

In the above task set, moon reduces gravity; tinyfeet shrinks the feet; inverted_actions applies the inverted action; nofeet removes control over several feet; and rainfall changes the friction coefficient, among other parameters. For the exact configurations, please refer to the original paper (Gaya et al., 2023).

