# OpenReview forum: "Neuro-evolutionary Continual Reinforcement Learning"
_ICML.cc/2026/Conference — ICML 2026 spotlight_

### Official Review · Reviewer_VKk8 · 2026-02-26

**Soundness:** 3
**Presentation:** 2
**Significance:** 2
**Originality:** 3
**Overall Recommendation:** 5
**Confidence:** 4

**Summary:**

Inspired by the human brain, the authors proposed a neuro-evolutionary continual reinforcement learning algorithm. The algorithm employs a fixed capacity network and uses evolutionary algorithm to expand masks which select larger subnetworks as more tasks arrive. Larger subnetworks are selected so that the additional neurons included introduce plasticity for learning the new task. Additionally, the authors proposed to freeze the parameters related to the old tasks to prevent catastrophic forgetting.

**Compliance With Llm Reviewing Policy:**

Affirmed.

**Final Justification:**

After considering the responses from the authors and the reviews from the other reviewers, I find the authors have addressed my concerns and work is sufficiently good, so I raised my score.

**Key Questions For Authors:**

Please address the weaknesses mentioned above.

**Limitations:**

The authors did not include a limitation section, I would recommend them to include one, which will be helpful for future readers.

**Strengths And Weaknesses:**

Strengths:

The authors analyzed the key components within the proposed algorithm.

The authors evaluated their algorithm against relevant existing algorithms such as PackNet and demonstrated their design to be more efficient under comparable "ratio of used network capacity".

Weaknesses:

The major weakness of the research is that the design is too complex. This has lead to two high level weaknesses:

(1) It is hard to evaluate what is the true cause for the Nevo-CRL method to has a better performance relative to relevant methods such as PackNet.

(2) Implementation details can be blurry. For example, I did not find descriptions on which version of network and mask should be used during testing.

With those weaknesses, it will be hard for future readers to understand the work and extract valuable insights from the work. At the same time, it will be hard for researcher to conduct further research on it.

Writing regarding the neural science motivation appears to be not very rigorous. For example, at around line 71 (left column), the authors claimed that since the human brain has a fixed capacity, so the policy network should also has a fixed capacity. It appears that the authors treats "imitating the human brain" as the ultimate solution to continual learning, which might mislead future readers.

---

> ### Author Rebuttal · Authors · 2026-03-31
>
> We appreciate the reviewer's valuable review and constructive comments, and we would like you to know that your questions provide considerably helpful guidance to improve the quality of our paper.
>
>
> 1. **[Re: what is the true cause for Nevo-CRL better than other methods.]**
>
> PackNet is a supervised continual learning method rather than a CRL method. It freezes parameters for previous tasks and improves parameter efficiency through pruning and retraining. CoTASP encodes task descriptions with Sentence-BERT, learns layer-wise dictionaries over task embeddings, and binarizes sparse prompts into task-specific masks.
>
> Nevo-CRL differs from these methods in several key aspects. Compared with CoTASP, it introduces semantic-similarity-based mask construction and population-based mask evolution, to enhance exploration. Its importance-based crossover and experience-sharing mechanisms further improve evolutionary and sample efficiency. Compared with PackNet, Nevo-CRL integrates pruning directly into the RL process and selectively reuses prior parameters, rather than reactivating all previously learned parameters.
>
> 2. **[Re: Implementation details can be blurry. which version of network and mask should be used during testing. ]**
>
> Regarding implementation, we will make the code publicly available once the paper is released, enabling timely reproduction of the experiments reported in the manuscript. We hope this addresses the reviewer’s concern.
>
> During testing, for the current task, since the best-performing individual cannot be determined a priori, we evaluate the policy associated with each mask and report the best performance among them. For previously learned tasks, we directly evaluate the best policy together with its corresponding mask.
>
> 3. **[Re: The neuroscience motivation seems insufficiently rigorous and potentially misleading.]**
>
> Thank you for the reviewer’s suggestion. We agree that the current wording is not sufficiently rigorous and may give readers the impression that our paper treats “imitating the human brain” as either the theoretical foundation or the ultimate solution for continual learning. In fact, the neuroscience-related motivation here is intended only as an intuitive source of inspiration, rather than a rigorous justification. In the revised version, we will tone down and clarify the relevant statements, and explicitly state that our method is only an exploratory attempt toward continual learning under a fixed-capacity setting. We also believe that other technical routes, such as experience replay and regularization-based methods, remain highly promising. A more reasonable long-term direction may be to combine the strengths of different approaches.
>
> 4. **[Re: I would recommend them to include limitation, which will be helpful for future readers.]**
>
> We sincerely thank the reviewer for the suggestion. We have discussed the limitations in the appendix and further summarize them here.
>
> - Additional time and computational overhead. Nevo-CRL maintains a population of policies, which inevitably increases computational cost. Although parallel training in JAX helps reduce this overhead, the overall training time is still about 50\% higher than that of non-population methods.
> - Inherited limitations of structure-based methods. Nevo-CRL inherits a common limitation of structure-based approaches: it currently lacks an effective mechanism for handling cases where network capacity is fully utilized. How to expand the network without affecting previously learned policies, or alternatively compress existing policies through parameter fusion, remains an important direction for future work.
> - Heuristic design. The current framework is largely heuristic and still lacks rigorous theoretical support and formal analysis.
>
> 5. **[More Experiments]**
>
> To further strengthen the empirical evaluation beyond the experiments on CW, we additionally conducted experiments across all Brax scenarios.
>
> Brax follows a design philosophy similar to CW and evaluates four aspects of continual learning: forgetting, transfer, robustness, and compositionality. It tests whether new tasks cause forgetting, whether prior learning hinders later tasks, whether policies remain stable under distractions, and whether prior skills can be reused for more complex tasks.
>
> The results are shown below:
>
> | Domain / Setting | CoTASP | Nevo-CRL |
> | --- | --- | --- |
> | Ant-Forgetting | 2746 | **4003** |
> | Ant-Transfer | 1315 | **3025** |
> | Ant-Robustness | 2446 | **4728** |
> | Ant-Compositional | 981 | **2200** |
> | Ant-Drastic-2 | 1615 | **3871** |
> | Ant-Drastic-1 | 2331 | **4134** |
> | Human-Official | 4192 | **4826** |
> | Human-Different Seq | 3385 | **3760** |
> | HalfCheetah-Forgetting | 1679 | **3341** |
> | HalfCheetah-Robustness | 4043 | **5672** |
> | HalfCheetah-Transfer | 781 | **3212** |
> | HalfCheetah-Composition | 3936 | **4278** |
>
> In all of these settings, Nevo-CRL consistently demonstrated significantly better performance than CoTASP.

---

> > ### Author Rebuttal · Reviewer_VKk8 · 2026-04-03
> >
> > After considering the responses from the authors and the reviews from the other reviewers, I find the work to be sufficiently good, so I will raise my score.

---

> > > ### Author Response · Authors · 2026-04-03
> > >
> > > We sincerely thank you for the valuable and in-depth comments, as well as for your recognition and support of our work, and we are especially grateful for your acknowledgment of our efforts in addressing your concerns.
> > >
> > >
> > > **We will ensure that all necessary clarifications and improvements are thoroughly and appropriately incorporated into the revised version.**
> > >
> > > **Thank you again for your time, thoughtful engagement, and kind support throughout the review process.**

---

### Official Review · Reviewer_hN23 · 2026-02-27

**Soundness:** 2
**Presentation:** 2
**Significance:** 2
**Originality:** 2
**Overall Recommendation:** 4
**Confidence:** 3

**Summary:**

The paper introduces a novel method for continual Reinforcement Learning: Neuro-evolutionary Continual Reinforcement Learning (Nevo-CRL). It is a fixed-capacity monolithic policy network that solves tasks by optimizing inter-layer connectivity and neuron parameters. The method is heavily brain-inspired, relies on pre-trained LLM embeddings for mask-similarity matching, and on evolutionary computation heuristics. The method is evaluated on the Continual World [1] benchmark against many baselines and yields strong performance, beating previous SoTA by a slight margin. Additional experiments conducted in BRAX are presented in the appendix. The paper ablates the method's design choices, indicating that each step contributes to its final performance.


[1] Wołczyk, M., Zajac, M., Pascanu, R., Kucinski, L., and Miłos, P. Continual world: A robotic benchmark for continual reinforcement learning. NeurIPS, 2021.

**Compliance With Llm Reviewing Policy:**

Affirmed.

**Final Justification:**

The rebuttal addressed my main concerns.

**Key Questions For Authors:**

Questions:
1. (General question) How scalable and are structure-based methods that work by applying masks to neurons. It seems this research direction isn't really used in practice, as far as I'm concerned.
2. How much more compute is used in Nevo-CRL in comparison to COTASP? It seems that population-based optimization in Nevo-CRL should introduce a non-negligible amount of additional compute.

Other comments:
1. Figure 1 is not helping in understanding the paper. I recommend improving the presentation of the method.
2. Figure 4 does not provide clear evidence that Nevo-RL works better than CoTASP. I suggest using a table to show efficient positive transfer.

**Limitations:**

yes

**Strengths And Weaknesses:**

Strengths:
1. The method seems to yield strong results on the Continual World benchmark and in Brax experiments, improving upon previous SoTA, COTASP [1]
2. There is an ablation of components of Nevo-CRL that clearly show how important LLM-created mask embeddings and evolutionary operations are as population-based optimization and crossover/pruning operations. Also, Nevo-CRL achieves the best performance while maintaining a low Ratio of Used Capacity.


Weaknesses:
1. The presentation of the method and results is poor. Figures are not improving the understanding of the paper. In particular, Figure 1 should be improved to make the method easier for readers to understand.
2. The paper does not provide a deeper analysis of why particular components of the method are important besides superficial ablations. As a result, the Nevo-CRL method is more of a bag of tricks that, when combined, yield some performance benefits in a continual RL setup. Yet this kind of evolutionary computing heuristic seems not to be scalable in more advanced settings, such as VLAs or LLMs with RLVR.
3. It's not clear how important it is to have a good language model for finding mask similarities.
4. While the method yields SoTA on the Continual World, it's not clear how scalable the method is.
5. The brain-inspired framing feels more like a post hoc justification of the method than a genuine attempt to understand the underlying principles.
6. The originality of the work is not well stated in Related Works.


[1] Yang, Y., Zhou, T., Jiang, J., Long, G., and Shi, Y. Continual task allocation in meta-policy network via sparse prompting. In ICML, 2023

---

> ### Author Rebuttal · Authors · 2026-03-31
>
> We will try our best to address each of the concerns and questions raised by the reviewer below:
>
> 1. **[Re: Figure 1 is not helping in understanding the paper.]**
>
> We sincerely thank the reviewer for this helpful suggestion. In the revision, we will further improve Figure 1 by streamlining the sparse mask construction workflow, presenting importance-based crossover as a unified module, and clarifying the population update process, including parameter update, mask replacement, and elite RL replacement.
>
> 2. **[Re: scalable in VLAs or LLMs with RLVR.]**
>
> Nevo-CRL has higher sample efficiency, making it particularly promising for transfer to VLAs, which is also a direction we are currently pursuing. A direct implementation is to retain a pretrained backbone and attach a lightweight decision head at the output layer, similar to RLT, followed by parameter adaptation through real-world RL or RL with sim2real. Under this setup, Nevo-CRL can leverage its experience-sharing mechanism and evolutionary search to construct multiple policies simultaneously for exploration and exploitation, thereby substantially reducing sample costs and improving its suitability for real-world VLA deployment.
>
> 3. **[Re: the importantance to have a good LM.]**
>
> We directly adopt the Sentence-BERT model used in CoTASP, as it is lightweight and sufficient for our setting. Even with imperfect representations, the population-based evolutionary search in Nevo-CRL can mitigate policy mismatch through candidate diversity.
>
> Stronger representation models, such as \texttt{text-embedding-3-large}, or LLM-based sampling and selection may further improve performance. Task descriptions could also be made more structured by using concise action-oriented phrases with object names and removing less informative words.
>
> 4. **[Re:  how scalable the method is.]**
>
> As a structure-based method, Nevo-CRL learns the same number of tasks with lower utilization (e.g., 56\% for 20 tasks) and better performance, indicating stronger scalability than CoTASP (76\%). When capacity is exhausted, expanding network width provides a natural extension: in the current four-layer architecture, scaling each layer by a factor of $n$ would theoretically increase capacity from $m$ to $m \times n^4$. In addition, the neuron overhead of new tasks gradually decreases as more tasks are learned, suggesting room for further scalability in practice.
>
> In addition, our experiments on all 12 Brax tasks further demonstrate the consistent superiority of Nevo-CRL in all HalfCheetah, Ant and Humanoid domain.
>
> | Domain | Relative Improvement (CoTASP = 1.0) |
> | --- | --- |
> | Ant | 2.02 |
> | Human | 1.13 |
> | HalfCheetah | 2.15 |
>
> 5. **[Re: The originality of the work.]**
>
> Nevo-CRL differs from previous methods in several key aspects. Compared with CoTASP, it introduces semantic-similarity-based mask construction and population-based mask evolution, to enhance exploration. Its importance-based crossover and experience-sharing mechanisms further improve evolutionary and sample efficiency. Compared with PackNet, Nevo-CRL integrates pruning directly into the RL process and selectively reuses prior parameters, rather than reactivating all previously learned parameters.
>
> 6. **[Re: the practicality of structure-based methods]**
>
> We believe structure-based methods are promising for real-world applications, especially embodied intelligence, which is our main direction for future work. Retraining the entire network for each new task is often impractical. Meanwhile, updating the network without degrading previously acquired skills remains highly challenging. Structure-based methods naturally address this need by enabling skill reuse with minimal interference. This is the main reason we focus on this line of research, though we do not exclude alternative solutions. We would also welcome further discussion from the reviewer.
>
> 7. **[Re: Compute overhead ]**
>
> Nevo-CRL introduces a policy population but adopts an experience-sharing architecture, where all experiences from different individuals are stored in a shared replay buffer. This improves sample efficiency while keeping the replay buffer size and network size unchanged relative to CoTASP.
>
> The main cost is additional computation. With a JAX implementation and static compilation, Nevo-CRL increases training time by 54\% over CoTASP while using about 2.6~GB of GPU memory on a single NVIDIA 4090 GPU.
>
> Overall, Nevo-CRL trades extra computation for higher sample efficiency, which is especially valuable for future real-world RL on robotic manipulators.
>
> 8. **[Re: Nevo-RL better than CoTASP.]**
>
> On CW10, Nevo-CRL reaches the two-time 80% threshold at 0.41M steps, compared to 0.47M for CoTASP. On CW20, it reaches 0.38M steps, versus 0.43M for CoTASP. In addition, we further evaluate on Brax, where Nevo-CRL achieves an average performance improvement of 1.96$\times$ over CoTASP across 12 tasks.
>
> ---
> **We are looking forward to more inspiring discussions.**

---

> > ### Author Rebuttal · Reviewer_hN23 · 2026-04-05
> >
> > Most concerns have been satisfactorily addressed. I will update my score.

---

> > > ### Author Response · Authors · 2026-04-05
> > >
> > > We are pleased to have addressed the reviewer’s concerns and sincerely appreciate your recognition and support of our work. The constructive suggestions are greatly helpful in improving the quality of our paper.
> > >
> > > We will carefully incorporate all manuscript refinements and experiments discussed into the revised version.
> > >
> > > In addition, we guarantee that the code will be made publicly available at the time of the paper’s release to ensure full reproducibility.
> > >
> > > Thank you again for your thoughtful feedback and the valuable discussions.

---

### Official Review · Reviewer_XcuM · 2026-03-08

**Soundness:** 3
**Presentation:** 3
**Significance:** 4
**Originality:** 3
**Overall Recommendation:** 4
**Confidence:** 2

**Summary:**

This paper proposes Nevo-CRL, a neuro-evolutionary framework for continual reinforcement learning. This paper considers a central aspect of mitigating catastrophic forgetting and plasticity loss through task-specific sparse masks on a fixed-capacity network. The authors proceed to consider a broad theme of integrating semantic similarity for mask initialization, population-based mask evolution, and importance-guided crossover to optimize connectivity. The method is evaluated on Continual World and Brax benchmarks, showing competitive results against existing baselines.

**Compliance With Llm Reviewing Policy:**

Affirmed.

**Key Questions For Authors:**

1.  How does semantic similarity correlate with actual policy transfer effectiveness? Could contradictory policies for similar tasks cause negative transfer?
2.  What is the computational overhead compared to single-network baselines (e.g., GPU hours)? Is the performance gain proportional to the cost?
3.  In long task sequences, how does the method handle capacity exhaustion when most neurons are frozen? Is there a mechanism to recycle old neurons beyond pruning?

**Limitations:**

Yes

**Strengths And Weaknesses:**

**Strengths:**
*  The integration of neuro-evolutionary principles with mask-based continual learning is novel. The importance-guided crossover and semantic mask construction distinguish this work from prior structure-based methods.
*  Experiments are comprehensive across multiple benchmarks (CW10, CW20, Brax) with strong baselines. Ablation studies support key design choices like population mechanism and pruning.
*   The paper is well-structured with clear figures and pseudocode. Related work is adequately covered.

**Weaknesses:**
*   Could the reliance on task descriptions for semantic similarity limit applicability when such information is unavailable or ambiguous? Might semantically similar tasks require contradictory policies, causing negative transfer?
*   Might the sensitivity to pruning thresholds (e.g., performance drops when threshold changes slightly) affect robustness across diverse domains?

---

> ### Author Rebuttal · Authors · 2026-03-31
>
> We appreciate the reviewer's valuable review and constructive comments, and we will try our best to address each of the concerns and questions raised by the reviewer below:
>
> 1. **[Re: Could the reliance on task descriptions for semantic similarity limit applicability when such information is unavailable or ambiguous? Might semantically similar tasks require contradictory policies, causing negative transfer?]**
>
> More accurate task semantics can improve knowledge utilization. Nevertheless, Nevo-CRL adopts a population-based strategy that samples historical masks according to similarity, thereby reducing the risk of suboptimality that may arise from reusing only the most similar mask.
>
> For settings in which tasks are semantically similar but require completely different policies, we have not yet identified a sufficiently representative benchmark example. However, we do provide evidence from Brax that supports this point from another perspective. In these Brax transfer scenarios, although tasks may appear related at the semantic level, the policy transferred from the previous task is actually detrimental to learning the new task, meaning that the capabilities required by the two tasks are effectively incompatible. Specifically, the transfer scenarios in Brax are designed such that a single policy has greater difficulty learning a new task after having learned the previous one than when learning the new task. In this setting, when the model starts learning the second task, the only available prior policy is precisely the one that hinders learning, making transfer between tasks inherently difficult.
>
> | Transfer | Ant | HalfCheetah |
> | --- | --- | --- |
> | CoTASP | 1315 | 781 |
> | Nevo-CRL | 3025 | 3212 |
>
> Despite this challenge, Nevo-CRL still yields significant improvements over CoTASP.
>
> 2. **[Re: Might the sensitivity to pruning thresholds (e.g., performance drops when threshold changes slightly) affect robustness across diverse domains?]**
>
> To address the reviewer’s concern, we perturb the pruning threshold by ±0.005 and report the results below:
>
> | CW20 | 0.045 | 0.05 | 0.055 |
> | --- | --- | --- | --- |
> | Nevo-CRL | 0.9 | 0.93 | 0.91 |
>
> The results show that slight perturbations to the pruning threshold cause some performance variation, while overall performance remains stable. In all Brax experiments, we use a fixed pruning threshold of 0.05. The results are shown below:
>
> | Domain / Setting | CoTASP | Nevo-CRL |
> | --- | --- | --- |
> | Ant-Forgetting | 2746 | **4003** |
> | Ant-Transfer | 1315 | **3025** |
> | Ant-Robustness | 2446 | **4728** |
> | Ant-Compositional | 981 | **2200** |
> | Ant-Drastic-2 | 1615 | **3871** |
> | Ant-Drastic-1 | 2331 | **4134** |
> | Human-Official | 4192 | **4826** |
> | Human-Different Seq | 3385 | **3760** |
> | HalfCheetah-Forgetting | 1679.11 | **3341.40** |
> | HalfCheetah-Robustness | 4043.43 | **5671.68** |
> | HalfCheetah-Transfer | 780.71 | **3211.93** |
> | HalfCheetah-Composition | 3936.48 | **4277.75** |
>
> Nevo-CRL consistently outperforms CoTASP across all 12 tasks, indicating the robustness of our hyperparameter settings.
>
> 3. **[Re: computational overhead compared to single-network baselines (e.g., GPU hours)? Is the performance gain proportional to the cost?]**
>
> Nevo-RL is implemented in JAX and uses `vmap` to accelerate population updates via multi-policy parallelism. Although this design improves efficiency, it still introduces additional computational overhead. Specifically, COTASP requires 45.9 hours to complete CW20, whereas Nevo-CRL requires 70.8 hours, representing a 54% increase in training time.
>
> Regarding memory overhead, the shared replay buffer prevents additional main memory consumption. In terms of GPU memory, the population training leads to a usage of approximately 2.6 GB on a single NVIDIA 4090 GPU (24 GB).
>
> The performance gains do not stem from these additional overheads. In contrast, Nevo-CRL achieves better performance under the same sampling budget, demonstrating higher sample efficiency. This property makes it particularly well suited to practical scenarios such as robotics, where data collection is costly and sample acquisition is expensive. In such settings, the advantage of Nevo-CRL becomes especially evident.
>
> ---
>
> We hope our replies have addressed the concerns the reviewer posed and shown the improved quality of the paper. **We are always willing to answer any of the reviewer's concerns about our work** and we are looking forward to more inspiring discussions.

---

> > ### Author Rebuttal · Reviewer_XcuM · 2026-04-03
> >
> > Most concerns have been satisfactorily addressed. However, I will maintain my original score.

---

> > > ### Author Response · Authors · 2026-04-04
> > >
> > > We are pleased to have addressed the reviewer’s concerns and sincerely appreciate your recognition and support of our work. The constructive suggestions are greatly helpful in improving the quality of our paper.
> > >
> > > We will carefully incorporate all manuscript refinements and experiments discussed into the revised version.
> > >
> > > **Thank you again for your time and insightful comments**.

---

### Official Review · Reviewer_2y9u · 2026-03-12

**Soundness:** 3
**Presentation:** 2
**Significance:** 2
**Originality:** 2
**Overall Recommendation:** 4
**Confidence:** 3

**Summary:**

The paper addresses the problem of continual reinforcement learning, where an agent must learn a sequence of tasks while retaining previously acquired knowledge and avoiding catastrophic forgetting. The authors propose Neuro-evolutionary Continual Reinforcement Learning (Nevo-CRL), a framework that maintains a fixed-capacity monolithic policy network and adapts it across tasks through neuro-evolutionary optimisation.

Instead of expanding the network or retraining separate models for each task, Nevo-CRL constructs a population of binary masks that selectively activate neurons across the layers of the policy network, effectively generating task-specific sub-policies within a shared architecture. For each new task, an evolutionary search process optimises the mask population to identify effective connectivity patterns. After training on a task, the best mask is stored and the neurons activated by that mask are frozen to protect previously learned knowledge and mitigate catastrophic forgetting.

To promote knowledge transfer across tasks, the framework encourages reuse of previously activated neurons while allocating unused capacity when necessary. The method is evaluated on a set of continual reinforcement learning benchmarks using Soft Actor-Critic as the underlying RL algorithm. Experimental results compare Nevo-CRL with several continual RL baselines, including regularisation-based, architecture-based, and rehearsal-based methods, using metrics such as average performance, forgetting, and generalisation. The results indicate that Nevo-CRL achieves improved performance and reduced forgetting while maintaining efficient network capacity usage.

**Compliance With Llm Reviewing Policy:**

Affirmed.

**Key Questions For Authors:**

1- The paper describes an evolutionary search over binary neuron masks while the policy parameters are trained with Soft Actor-Critic. Could the authors clarify how the evolutionary mask optimisation interacts with policy learning during training (e.g., timing, update frequency, and whether policies are fully retrained or partially reused for each mask)? A clearer description of this process would improve reproducibility and help assess whether the reported gains arise from the masking strategy itself or from additional optimisation steps. A detailed explanation could strengthen confidence in the technical soundness of the method.

2- The proposed framework introduces an evolutionary search over a population of masks in addition to standard reinforcement learning updates. Could the authors provide a quantitative analysis of the computational overhead compared to the baselines, including training time and resource usage? Understanding the cost and scalability of the method, especially as the number of tasks increases, would help evaluate the practical significance of the approach.

3- The method freezes neurons associated with previously learned tasks to prevent forgetting. How does the approach behave when the number of tasks grows and the available network capacity becomes limited? It would be helpful to understand whether the framework includes mechanisms for capacity management or neuron reuse, and how performance degrades when the network approaches full utilisation. Clarification on this point could affect the assessment of the method’s applicability to long task sequences.

4- The framework combines several elements, including evolutionary mask search, neuron freezing, and neuron reuse. Could the authors provide ablation studies that isolate the contribution of these components to the overall performance? Such analysis would clarify which elements are most responsible for the reported improvements and would strengthen the empirical evidence supporting the proposed design.

**Limitations:**

Yes

**Strengths And Weaknesses:**

The paper addresses the problem of continual reinforcement learning, where an agent must learn a sequence of tasks while avoiding catastrophic forgetting. The proposed Nevo-CRL framework introduces a neuro-evolutionary mechanism that searches for task-specific subnetworks within a fixed-capacity monolithic policy network by evolving binary neuron masks. This approach allows the model to allocate subsets of neurons to different tasks while freezing previously used neurons to preserve past knowledge. The method is integrated with Soft Actor-Critic and evaluated on several continual reinforcement learning benchmarks. The experimental results suggest that the proposed approach can mitigate forgetting while maintaining competitive performance. The paper tackles a relevant problem in reinforcement learning and presents an interesting combination of ideas from neuro-evolution, subnetwork selection, and continual learning.

However, the novelty relative to existing approaches is somewhat limited, as subnetwork masking and neuron isolation strategies have been explored in prior continual learning work, and evolutionary optimisation for architecture or connectivity search is also well established. The paper would benefit from a clearer discussion of how the proposed method differs from or advances these existing approaches. Some methodological aspects are also insufficiently analysed. In particular, the evolutionary search introduces additional computational cost and hyperparameters, but the scalability and efficiency of the approach are not discussed in detail. The experimental evaluation is reasonable but somewhat limited, and additional ablation studies would help clarify the contribution of key components such as mask evolution and neuron freezing. While the presentation is generally clear, the description of the evolutionary mask optimisation and its interaction with policy learning could be more precise to improve reproducibility. Overall, the work addresses an important problem and proposes an interesting combination of techniques, but further clarification and deeper empirical analysis would strengthen the contribution.

---

> ### Author Rebuttal · Authors · 2026-03-31
>
> We appreciate the reviewer's valuable review and constructive comments, we will try our best to address each of the concerns and questions raised by the reviewer below:
>
> 1. **[Re: Could the authors provide ablations for evolutionary mask search, neuron freezing, and neuron reuse?]**
>
> To address the reviewer’s concern, we conduct a series of ablation studies. **The results show that evolutionary mask search, neuron freezing, and neuron reuse each contribute positively to overall performance**.
>
> Specifically, evolutionary mask search helps the model identify high-quality masks more efficiently. Removing neuron freezing leads to severe performance collapse, indicating that the framework can no longer preserve performance on previously learned tasks. Removing neuron reuse causes neuron utilization to reach 100% by around Tasks 10–11, after which the model can no longer accommodate additional tasks.
>
> | CW20 | Nevo-CRL | w/o Evolutionary Mask Search | w/o Neuron Freezing | w/o Neuron Reuse |
> | --- | --- | --- | --- | --- |
> | Peformance | 0.93 | 0.87 | 0.14 | 0.45 |
> | Capacity Utilization Rate | 56% | 56% | - | 100% |
>
> 2. **[Re: How does the method scale under increasing tasks and limited capacity?]**
>
> Nevo-CRL completes 20 tasks with a four-layer network of 1024 neurons per layer, using only 56\% of the available neurons on average. Compared with CoTASP (76\%), this indicates higher neuron utilization efficiency and better scalability among structure-based methods.
>
> In principle, task capacity can be further increased by widening each layer. For the current four-layer architecture, which supports up to $m$ tasks, scaling each layer by a factor of $n$ would theoretically increase task capacity to $m \times n^4$. Moreover, we observe that the neuron overhead of each new task gradually decreases as more tasks are learned, suggesting the potential to support even more tasks in practice.
>
> Once the network reaches its capacity limit, one possible solution is to add new neurons to each layer while disconnecting them from existing neurons, thereby preserving previously learned tasks. We view this as a promising direction for future work.
>
> 3. **[Re: A clearer discussion of how the proposed method differs from or advances these existing approaches.]**
>
> Thank you for your suggestion.
>
> PackNet is a supervised continual learning method rather than a CRL method. It freezes parameters for previous tasks and improves parameter efficiency through pruning and retraining. CoTASP encodes task descriptions with Sentence-BERT, learns layer-wise dictionaries over task embeddings, and binarizes sparse prompts into task-specific masks.
>
> Nevo-CRL differs from these methods in several key aspects. Compared with CoTASP, it introduces semantic-similarity-based mask construction and population-based mask evolution, to enhance exploration. Its importance-based crossover and experience-sharing mechanisms further improve evolutionary and sample efficiency. Compared with PackNet, Nevo-CRL integrates pruning directly into the RL process and selectively reuses prior parameters, rather than reactivating all previously learned parameters.
>
> We will add this discussion in the revised version.
>
> 4. **[Re: clarify how the evolutionary mask optimisation interacts with policy learning during training]**
>
> The mask is updated once every 200,000 environment steps, which means it is updated five times within 1 million environment steps for each task.
>
> During mask evolution, each new mask is generated by crossover between parent masks, and the corresponding policy parameters are directly inherited from the parents. Specifically, if a unit in the new mask comes from mask A, it inherits the corresponding parameter from policy A; if it comes from mask B, it inherits the corresponding parameter from policy B.
>
> 5. **[Re: A clearer description of this process would improve reproducibility.]**
>
> Thank you for your suggestion. We will improve this part based on your suggestion in the revised version.
>
> Meanwhile, we will make the code publicly available upon publication to ensure that the proposed method can be readily reproduced.
>
> 6. **[Re: Provide a quantitative analysis of training time and resource usage. ]**
>
> Nevo-RL is implemented in JAX and uses `vmap` to accelerate population updates via multi-policy parallelism. Although this design improves efficiency, it still introduces additional computational overhead. Specifically, COTASP requires 45.9 hours to complete 20 tasks in CW20, whereas Nevo-CRL requires 70.8 hours, representing a 54% increase in training time.
>
> Regarding memory overhead, the shared replay buffer prevents additional main memory consumption. In terms of GPU memory, the population training leads to a usage of approximately 2.6 GB on a single NVIDIA 4090 GPU (24 GB).

---

> > ### Author Rebuttal · Reviewer_2y9u · 2026-04-02
> >
> > I thank authors for their comments. Their response is satisfactory. However,  I would like to keep my scores.

---

> > > ### Author Response · Authors · 2026-04-02
> > >
> > > We are pleased to have addressed the reviewer’s concerns and **sincerely appreciate your recognition and support of our work. The constructive suggestions are greatly helpful in improving the quality of our paper.**
> > >
> > > We will carefully incorporate all manuscript refinements and experiments discussed into the revised version.
> > >
> > > In addition, we would like to note that, beyond the CW10 and CW20 experiments reported in the main text, we have also completed the evaluation and verification on all tasks in Brax.  Below, we first provide a detailed introduction to several tasks in Brax. Brax is inspired by the design philosophy of CW and constructs task sequences targeting different skill dimensions.
> > >
> > > The followings are some common **task sequence scenarios** in Brax:
> > >
> > > 1. **Forgetting Scenarios** are designed such that a single policy tends to forget the former task when learning a new one.
> > >
> > >
> > > 2. **Transfer Scenarios** are designed such that a single policy has more difficulties to learn a new task after having learned the former one, rather than learning it from scratch.
> > >
> > > 3. **Robustness Scenarios** alternate between a normal task and a very different distraction task that disturbs the whole learning process of a single policy (simply inverted the actions). In these tasks, fine-tuning policies struggle to recover good performances (the final average reward actually decreases).
> > >
> > > 4. **Compositional Scenarios** present two first tasks that will be useful to learn the last one, but a very different distraction task is put at the third place to disturb this forward transfer. The last task is indeed a combination of the two first tasks in the sense that it combines their particularities.
> > >
> > > Both HalfCheetah and Ant include the above scenarios.
> > >
> > > The details in Ant domain (Each sequence is repeated once, with 1 million steps per task)
> > >
> > > Forgetting: normal → hugefeet → rainfall → moon
> > >
> > > Transfer: nofeet_1_3 → nofeet_2_4 → nofeet_1_2 → nofeet_3_4
> > >
> > > Robustness: normal → inverted_actions → normal → inverted_actions
> > >
> > > Compositional: nofeet_2_3_4 → nofeet_1_3_4 → nofeet_1_2 → nofeet_3_4
> > >
> > > Drastic change 1: hugefeet → moon → rainfall → nofeet_1_3
> > >
> > > Drastic change 2: nofeet_2_3_4 → moon → inverted_actions → hugefeet
> > >
> > > Humanoid domain with higher dimensions,
> > >
> > > Official version: normal → moon → carrystuff → tinyfeet (No repetition with 2 million steps per task)
> > >
> > > Different seq: tinyfeet → moon → carrystuff → normal (Each sequence is repeated once, with 1 million steps per task) In the above task set, **moon** reduces gravity; **tinyfeet** shrinks the feet; **inverted_actions** applys the **inverted** action; **nofeet** removes control over several feet; and **rainfall** changes the friction coefficient, among other parameters. For the exact configurations, please refer to the original paper[1].
> > >
> > > | Ant Domain | Forgetting | Transfer | Robustness | Compositional | Drastic change 2 | Drastic change 1 |
> > > | --- | --- | --- | --- | --- | --- | --- |
> > > | CoTASP | 2746 | 1315 | 2446 | 981 | 1615 | 2331 |
> > > | Nevo-CRL | **4003** | **3025** | **4728** | **2200** | **3871** | **4134** |
> > >
> > > | Human Domain | Official | Different Seq |
> > > | --- | --- | --- |
> > > | CoTASP | 4192 | 3385 |
> > > | Nevo-CRL | **4826** | **3760** |
> > >
> > > | HalfCheetah | Forgetting | Robustness | Transfer | Composition |
> > > | --- | --- | --- | --- | --- |
> > > | CoTASP | 1679.11 | 4043.43 | 780.71 | 3936.48 |
> > > | Nevo-CRL | **3341.40** | **5671.68** | **3211.93** | **4277.75** |
> > >
> > > Including the CW10 and CW20 experiments, we have evaluated a total of 14 task sequences. **The results show that Nevo-CRL consistently outperforms CoTASP (the current state-of-the-art structured method)** on all task sequences.
> > >
> > > [1]. Building a Subspace of Policies for Scalable Continual Learning

---

### Decision · Program_Chairs · 2026-04-30

**Decision:**

Accept (spotlight)

**Comment:**

The paper addresses an important problem in continual reinforcement learning and proposes a technically solid method with strong empirical results. Reviewers found the approach promising, and the reported gains over relevant baselines, together with the additional ablations and clarifications provided during the rebuttal period, give sufficient support for the paper’s main claims.

Reviewers raised concerns about novelty relative to prior structure-based methods, the complexity of the design, scalability and computational overhead, and aspects of presentation and clarity. After the rebuttal, however, the reviewers agreed that these issues were sufficiently addressed or are better viewed as limitations than as rejection-level flaws. Overall, the strengths outweigh the remaining weaknesses.